# Associations Between Thoracic Ultrasound Chute-Side Evaluations and 60-Day Outcomes in Feedyard Cattle at Time of First Treatment for Respiratory Disease

**DOI:** 10.3390/vetsci12040369

**Published:** 2025-04-15

**Authors:** Luis F. B. B. Feitoza, Brad J. White, Robert L. Larson, Tyler J. Spore

**Affiliations:** 1Beef Cattle Institute, Kansas State University, Manhattan, KS 66506, USA; lffeitoza@vet.k-state.edu (L.F.B.B.F.); rlarson@vet.k-state.edu (R.L.L.); 2Innovative Livestock Services, 2006 Broadway Ave., Great Bend, KS 67530, USA; tyler.spore@ilsbeef.com

**Keywords:** point-of-care diagnostics, feedlot cattle health, precision veterinary medicine, bovine respiratory disease, prognostics

## Abstract

When cattle first show signs of pneumonia, it is vital for cattle caretakers and veterinarians to quickly determine which animals are at higher risk of not recovering. In this study, we tested a small ultrasound device that can be used while the animals are still in the chute. By examining the lungs and gathering basic information—like weight and how many days the animal has been in the feedyard—researchers identified which cattle were more likely to need extra treatments or be removed from the herd (for example, due to severe illness). Cattle with more lung abnormalities observed by thoracic ultrasonography were more often the ones that did not respond well to initial treatment. This information helps cattle producers and veterinarians provide more targeted care, improving animal welfare and saving resources. Early and accurate predictions of illness severity also allow for better management decisions. Overall, these ultrasound-based measurements can help assess the severity of the respiratory disease and aid animal caretakers in making more informed management decisions.

## 1. Introduction

Bovine respiratory disease (BRD) is a significant concern in the feedyard industry, representing one of the leading causes of morbidity and mortality in cattle [1,2]. The economic implications of BRD are substantial, encompassing not only direct costs related to treatment and mortality, but also indirect costs resulting from reduced weight gain, poor carcass quality, and overall productivity [2,3,4,5]. The complexity of BRD in cattle can often be attributed to a combination of viral and bacterial pathogens, environmental stressors, and management practices, making diagnosis and prognosis difficult [1,6,7]. Beyond economic impacts, BRD-affected cattle often experience severe distress and prolonged recovery periods, which is a significant animal welfare concern [2].

Accurately prognosing cattle at the time of first BRD treatment is a critical yet challenging aspect BRD management. Early and accurate prognosis is essential for informing management practices, guiding decisions on treatment interventions, and determining the appropriate allocation of resources [8,9]. Potentially, cattle welfare can be positively impacted by these decisions, as an accurate prognosis can prevent unnecessary suffering by identifying animals that may not benefit from further treatment, but will benefit from early culling [10].

Moreover, point-of-care (POC) diagnostics have potential to be adopted as valuable tool for enhancing prognostic accuracy at the chute-side. POC tools such as pulse oximetry [11,12,13], lung auscultation [14,15], and thoracic ultrasound have been tested and utilized in veterinary medicine to provide real-time insights into the pulmonary health of cattle [11,16,17,18,19,20,21]. These tools can be particularly relevant in feedyard operations, where rapid and accurate assessments are crucial for effective decision-making. Thoracic ultrasound, for instance, allows for the visualization of pulmonary abnormalities that may not be detectable through traditional methods [22], while pulse oximetry provides a non-invasive measure of blood oxygen saturation [11], which can offer critical information about respiratory function.

Despite the potential benefits of POC diagnostics, there is a notable gap in the literature regarding their use for prognosing first treatment failure (FTF) and death/culling (DNF) in feedyard cattle at time of treatment [23]. Existing research has primarily focused on the application of these tools in clinical settings, with limited exploration of their utility in chute-side scenarios. This lack of data underscores the need for studies that evaluate the associations between chute-side diagnostic parameters and critical outcomes such as FTF and DNF.

The objective of this study was to address this gap by assessing the associations between specific chute-side parameters, including cattle demographics, targeted thoracic ultrasonography, pulmonary auscultation, and pulse oximetry, with the association with first treatment failure and not finishing the feeding phase (DNF) in a 60-day post-enrollment interval. By identifying valuable prognostic parameters for use at the chute-side, this research aims to contribute to the broader goal of improving feedyard cattle management, ultimately enhancing both animal welfare and economic sustainability in the industry.

## 2. Materials and Methods

### 2.1. Experimental Design and Enrollment Criteria

This research was designed as a cross-sectional observational study, with individual feedyard cattle serving as the experimental units. The study aimed to evaluate pulmonary health at the time of first treatment for bovine respiratory disease (BRD) using various point-of-care (POC) diagnostic tools. Data collection occurred from summer to fall of 2023 at a commercial cattle operation located in the high plains of the United States and was performed by a trained veterinarian (LF).

Enrolled cattle were selected based on their first treatment for respiratory disease. Animals were selected from home pens to be treated and were treated by feedyard personnel. None of the POC evaluations or data were shared with feedyard personnel at the time of treatment to avoid influencing their management decisions.

Demographic data, including sex, breed, bodyweight (BW), and days on feed (DOF), were recorded for each animal at time of treatment. The study focused on identifying associations between POC diagnostic parameters and 60-day post-enrollment outcomes: first treatment failure (FTF) and did not finish (DNF). First treatment failure refers to an animal that was re-treated, was culled, or died before the 60-day post-enrollment marker. The criteria for any treatment intervention (1st or other) was decided by the feedyard’s herd health personnel. The study investigators were not engaged in this and had no impact on their decision regarding the drug of choice or decision to treat an animal. DNF refers to animals that did not complete the feeding phase due to mortality or culling. Cattle that were deemed finished (decided by the commercial operation) and were sent to the abattoir before the end of the 60-day evaluation period were excluded from the analysis.

Chute-side point-of-care evaluation parameters were collected by the use of pulse oximetry technology, which is capable of measuring blood oxygen saturation (SPO2) and pulse per minute (PPM); lung auscultation targeted two thoracic locations on the right lung only (cranioventral and caudo-dorsal); and the last was targeted thoracic ultrasound (TT-POCUS) at the caudo-dorsal level. This evaluation was also carried only on the right lung. Pulse oximetry and pulmonary auscultation were evaluated on a subset of 443 cattle due to a protocol deviation where these parameters were late to be added to the study, and were added after 376 cattle were enrolled already.

### 2.2. Pulse Oximetry

Pulse oximetry was performed by a trained veterinarian (LF) on a subset of cattle (n = 443), using a pulse oximeter unit equipped with a transflectance sensor. The sensor was placed on the external ear (between the cartilage ribs) after cleaning to remove debris, and data were recorded once the signal stabilized, which took around 15 s. The measurements included blood oxygen saturation (SPO2) and pulse per minute (PPM).

### 2.3. Pulmonary Auscultation Score

Pulmonary auscultation was conducted on the same subset of cattle (n = 443), with evaluations performed for less than 30 s per animal. The stethoscope was placed on the right caudo-dorsal lung lobe and the cranio-ventral lung field. Specifically, the cranio-ventral assessment was conducted between the 4th and 5th ribs caudal to the olecranon (elbow), while the caudo-dorsal assessment took place on the 8th intercostal space (ICS). Pulmonary auscultation scores (PASs) were assigned on a scale from 1 to 5, based on the presence and severity of abnormal lung sounds, following a modified scoring system derived from the work of DeDonder et al. (2010) [15]. Score was defined as follows: 1—normal lung sounds; 2—presence of mild crackles/rales; 3—presence of moderate crackles/rales; 4—presence of severe crackles/rales; 5—presence of severe diffuse crackles/rales.

### 2.4. Targeted Thoracic Point-of-Care Ultrasound

A targeted thoracic ultrasonography (TT-POCUS) procedure was performed on all enrolled cattle (n = 819) while they were restrained in a hydraulic squeeze chute. The ultrasound device of choice had a frequency range of 1 MHz to 10 MHz and was equipped with a specialized POC probe featuring a 5 cm × 3 cm footprint, with an optimized preset (lung preset) for lung imaging and usage over hair coat, not requiring shaving or trimming. Isopropyl alcohol 70% was utilized to enhance probe-to-skin contact. Each ultrasound examination was limited to a 60 s duration to accommodate the feedyard operation routine. The ultrasound area of interest was in the thoracic region correspondent to the right caudo-dorsal lung lobe. The ultrasound probe was placed in a short-axis orientation in the ICS between the 8th and 11th ribs on the right side to scan all areas of interest. The “fanning” technique (movement of the probe in a side-to-side fashion) was applied to maximize the area scanned and enhance the visualization of lung tissue.

Ultrasound lung scores (ULSs) were assigned based on the observed lung abnormalities, and were categorized as follows: score 1—fewer than 3 thin (<7 mm) B-lines; score 2—3 or more thin (<7 mm) B-lines; score 3—merged (>7 mm) B-lines; score 4—multiple wide/merged (>7 mm) B-lines with abnormal pleural findings (e.g., moth sign); score 5—consolidated lung with pleural thickening/irregularities and effusion. In addition, the B-line count category (0 to 2 and ≥3), A-line count category (0 to 2 and ≥3), and the presence of pleural abnormalities were recorded (moth sign). Post hoc analysis included measurement of the B-line area (cm^2^) using ImageJ 1.54g software from a still frame from the 60 s clip, where the image of choice was the frame with the most pulmonary abnormalities observed. This post hoc measurement was carried on a sub-sample of 710 cases out of the 819 total collected for this study due to availability to saved ultrasound files.

### 2.5. Statistical Analysis

Data management and statistical analyses were conducted using Microsoft Excel and RStudio 12.1g software. Descriptive statistics were summarized using the “dplyr” and “tidyr” packages in RStudio. The variables, including bodyweight, days-on-feed, B-line count, and A-line count, were categorized for analysis. The BW was categorized as <272, 272 to 361, 362 to 453, and >453 kg. Days-on-feed (DOF) were categorized based on recent published data [24]. Feeding phases were epidemiologically categorized as early-phase (0 to 42 days), mid-phase (43 to 71 days), and late-phase (>71 days) DOF based on the incidence of respiratory disease for each feeding phase. B-line count was categorized as <3, and ≥3, based on established thresholds indicative of pulmonary injury [25]. The A-line count was categorized as <3 and ≥3; the rationale behind this cutoff was of an exploratory nature. The other categorical variables included sex (steer and heifers), ULS (1–5), PAS (1–5), and moth sign (yes or no). Continuous data included SPO2, pulse rate, and B-line area (cm^2^).

Generalized linear mixed-effects models were implemented using the “glmer” function from the “lme4” package in RStudio, with a binomial response variable (logit function). The model included fixed effects of sex, BW and DOF, and random effects of cattle lot. Stepwise model selection was conducted using the “stepAIC” function from the “MASS” package, with variables removed based on the least significance and model fitness using AIC values. Multicollinearity was assessed using variance inflation factors (VIFs), with a VIF > 2.5 considered indicative of high collinearity [26]. The significance level was set at *p* < 0.05.

Two outcomes of interest, DNF (cull/dead) and FTF (re-treat/cull/dead), were tested to measure their association with POC evaluation parameters data. Each model was tested separately, and model-adjusted probabilities were calculated based on the final model for each outcome of interest. Following model estimation, the log-odds were transformed into probabilities to facilitate interpretation. Transformation was performed using the “emmeans” package in RStudio, enabling the computation of probabilities for each dependent variable included in the model.

## 3. Results

### 3.1. Descriptive Statistics

#### 3.1.1. Did Not Finish Model

At the time of the first respiratory treatment event, 819 cattle were POC evaluated. Of those, 474 were heifers (58%) and 345 were steers (42%) (Table 1). Overall, 70% of cases were in the early feeding phase (0 to 42 DOF; n = 573) category, 13% of cases were in the mid-feeding phase (43 to 71 DOF; n = 104) category, and 17% cases were in the late feeding phase (>71 DOF; n = 142) category. Seventy-five percent of cases were within the 272 to 361 kg (n = 392) and 362 to 453 kg (n = 224) weight categories. Within the 60-day post-evaluation period, most cases were classified as recovered after the first treatment (77%; n = 628) and 23% (n = 191) cases were culled or died, accounting for the inherent basal prevalence of DNF in this studied population.

The distribution of cases enrolled by DOF at the time of first treatment grouped by treatment success (60-day outcome) indicates that DNF cases occurred later on the feeding phase and were dispersed throughout the feeding period (median of 48 DOF and interquartile range (IQR) 91 DOF). In contrast, the recovered cases display an apparent narrower distribution, with a median of 21 DOF and IQR of 33 DOF (Figure 1).

Assessment by TT-POCUS revealed that most cases (82.4%; n = 675/819) were classified as ULS 1, 2, or 3. The majority of the cases that displayed 0–2 B-line count (482, 88%) recovered after the first treatment. Most cases also had ≥3 A-lines (459, 56%), had no effusion (664, 81%), and did not present moth signs (569, 69%). The B-line area average was 19.7 ± 18.2 cm^2^ for cattle that DNF and 16.11 ± 13.1 cm^2^ for cattle that recovered. Blood oxygen saturation averaged 83.1% ± 9.4% for cattle that DNF and 85.7% ± 8.5% for cattle that recovered. The pulse averaged 75.9 ± 25 per minute for cattle that DNF and 74.5 ± 22 PPM for cattle that recovered. Cranioventral PAS indicated that 92% of cattle given scores 1, 2, and 3 recovered, whereas 24.2% of cattle with scores 4 and 5 did not finish (DNF) treatment. Nevertheless, only 35% of cattle that presented a cranioventral PAS of 5 did not finish. When caudo-dorsal PAS was considered, 9.5% of cattle received scores of 1, 2, and 3 DNF, while 50% of cattle FTS with PAS 5.

#### 3.1.2. First Treatment Failure Model

At the time of the first respiratory treatment event, 819 cattle were POC evaluated, although only 443 cattle were evaluated with pulse oximetry and pulmonary auscultation due to a protocol deviation where these parameters were added to the study after 376 cattle were already enrolled. There were 474 heifers (58%) and 345 steers (42%) enrolled with the descriptive statistics listed in Table 2. There were 573 cases (70%) in the early feeding phase (0 to 42 DOF) category, 104 cases (13%) in the mid feeding phase (43 to 71 DOF) category, and 142 cases (17%) in the late feeding phase (>71 DOF) category. Seventy-five percent of the 819 cases were within the 272 to 361 kg and 362 to 453 kg categories. Most cases obtained success after the first treatment (n = 492, 60%) within the 60-day post-evaluation period and 327 cases exhibited first treatment failure (40%, inherent basal prevalence of FTF in this studied population).

The distribution of cases enrolled by DOF at time of first treatment grouped by treatment success at 60-day outcome indicates that the occurrence of FTF cases was dispersed throughout the feeding period (median of 28 DOF and interquartile range (IQR) 57 DOF). In contrast, recovered cases displays an apparent narrower distribution with a median of 22 DOF and IQR 36 DOF (Figure 2).

The descriptive statistics for POC measures in the treatment success model are listed in Table 2. Assessment by TT-POCUS revealed that most cases (688/819, 84%) were given ULSs of 2, 3, or 4. The majority of the cases that displayed three or more B-line count (156, 58%) resulted in treatment failure. Most cases also had ≥3 A-lines (459, 56%), lacked effusion (664, 81%), and did not present moth signs (569, 69%). The B-line area average for DNF was 19.7 ± 18.2 cm^2^ and 16.11 ± 13.1 cm^2^ for the recovered outcome. Blood oxygen saturation (%) averaged 85.9% ± 8.3% for FTS cattle and FTF cases averaged 84.1% ± 9.2%. Pulse (PPM) averaged 74.6 ± 22 PPM for FTS cattle and 75.1 ± 23 PPM for FTF. Cranioventral PAS indicated that 87.4% of cattle with scores of 1, 2, and 3 displayed first treatment success, whereas 43.2% of cattle with scores of 4 and 5 presented treatment failure; 51% of cattle with cranioventral PAS of 5 had a first treatment failure outcome. Caudo-dorsal PAS indicated that 29% of cattle placed in scores 1, 2, and 3 failed the first treatment, while 49.5% of cattle FTS with PASs of 4 and 5.

### 3.2. Logistic Regression

#### 3.2.1. Did Not Finish Model

The DNF multivariate logistic regression detected evidence of five variables to be associated with DNF; two variables related to cattle demographics, DOF and BW (*p* = 0.001, and *p* = 0.007, respectively); and three related to TT-POCUS, B-line count category, moth sign, and ULS (*p* = 0.001, *p* = 0.001, and *p* = 0.04, respectively). Although it was included in the model as a fixed variable, sex was not significantly associated with DNF (*p* = 0.22). In contrast, the variables that were not significantly associated with DNF outcome and therefore were not selected by the study’s multivariate logistic regression were A-line category, presence of pleural effusion, B-line area, SPO2, pulse rate, and both PASs (*p* > 0.05). No variables needed to be removed from the final model due to collinearity. The VIFs were 1.1 (sex), 1.9 (DOF), 1.5 (ULS), 2.0 (BW), 1.3 (B-line count), and 1.2 (moth sign).

In Table 3, Tukey-adjusted probabilities obtained using the fitted model showed that cases that received first treatment for respiratory disease mid DOF (43 to 71 DOF) or late DOF (>71 DOF) were more likely to DNF (32 ± 7%, and 64 ± 6%, respectively). In contrast, cattle that were treated for respiratory disease for the first time early in the feeding phase (0 to 42 DOF) were less likely to DNF (18 ± 3%). The probabilities of DNF by BW showed that cattle <272 kg and >453 kg were more likely to DNF (61 ± 10% and 48 ± 8%, respectively), while 272 to 361 kg (34 ± 6%) and 362 to 453 kg (25 ± 5%) were less likely to DNF. The probability of DNF was higher when B-line ≥ 3 (60 ± 6%) compared to when B-line 0 to 2 (24 ± 4%). The presence of moth signs was associated with a higher probability of DNF (51 ± 5%) compared to when moth sign was absent (24 ± 4%). The probabilities of DNF in association with ULSs are displayed in Figure 3. The probabilities of ULSs of 1 (18 ± 5%), 2 (24 ± 4%), 3 (28 ± 4%), and 4 (38 ± 6%) were not different amongst them (*p* > 0.05). However, ULS 5 (74 ± 8%) showed evidence of significant difference from all other scores (*p* = 0.001), showing the greatest probability of DNF.

#### 3.2.2. First Treatment Failure Model

The FTF multivariate logistic regression also detected evidence of five variables to be associated with FTF, DOF and BW (*p* = 0.007, and *p* = 0.01, respectively), and three TT-POCUS variables: B-line count category, moth sign, and ULS (*p* = 0.001, *p* = 0.001, and *p* = 0.03, respectively). Although it was included in the model as fixed variable, sex was not significantly associated with DNF (*p* = 0.23). In contrast, variables that were not significantly associated with FTF outcome and therefore were not selected by the study’s multivariate logistic regression were A-line category, presence of pleural effusion, B-line area, SPO2, pulse rate, and both PASs (*p* > 0.05). No variables were removed from the final model due to multicollinearity. The VIFs were 1.0 (sex), 1.8 (DOF), 1.3 (ULS), 1.9 (BW), 1.2 (B-line count), and 1.1 (moth sign).

In Table 3, Tukey-adjusted probabilities obtained using the fitted model showed that cases that received their first treatment for respiratory disease late in the feeding phase (>71 DOF) were more likely to exhibit FTF (66 ± 6%). In contrast, cattle that were treated for respiratory disease for the first time early on the feeding phase (0 to 42 DOF) to mid DOF (43 to 71 DOF) were less likely to exhibit FTF (43 ± 4%, and 44 ± 7%, respectively). Probabilities of FTF by BW showed that cattle weighing <272 kg and >453 kg were more likely to exhibit FTF (60 ± 8% and 53 ± 6%, respectively), while those weighing 272 to 362 kg (53 ± 5%) and 362 to 453 kg (38 ± 5%) were less likely to exhibit FTF. The probability of FTF was higher when the number of B-lines was ≥ 3 (64 ± 5%) compared to when the number of B-lines was 0 to 2 (43 ± 4%). The presence of moth sign was associated with a higher probability of DNF (65 ± 4%) compared to when moth sign was absent (37 ± 4%). The probabilities of FTF by ULS are displayed in Figure 4. The probabilities of ULSs of 1 (36 ± 6%), 2 (37 ± 4%), and 3 (41 ± 4%) did not differ (*p* > 0.05). However, 4 (57 ± 6%) and ULS 5 (75 ± 7%) showed evidence of significant difference from all other scores (*p* = 0.006), showing a greater probability of FTF.

## 4. Discussion

This study aimed to investigate the potential prognostic value of various chute-side diagnostic tools, specifically targeted thoracic ultrasonography, pulse oximetry, and pulmonary auscultation, in predicting the outcomes of first treatment failure and unfinished treatment in feedyard cattle with bovine respiratory disease at the point of the first treatment. The findings shows that certain chute-side diagnostic parameters, such as TT-POCUS variables, are significantly associated with the likelihood of these negative outcomes. These associations offer valuable information for improving the prognosis of cattle treated for BRD and will contribute to better management practices.

### 4.1. Associations with First Treatment Failure and Did Not Finish Outcomes

The logistic regression models identified significant associations between both FTF and DNF outcomes and key parameters, such as days on feed (DOF), bodyweight (BW), B-line count category, ultrasound lung score (ULS), and the presence of the moth sign. The ULS and B-line count emerged as particularly important variables in both models, underscoring the value of TT-POCUS as a prognostic tool for BRD. Specifically, cattle with ULSs of 5, indicating more severe lung pathology, had a significantly higher probability of FTF and DNF compared to those with lower ULSs.

### 4.2. Days on Feed and Bodyweight as Prognostic Indicators

The association between DOF and BW and both FTF and DNF outcomes highlights the influence of feeding stage and cattle size on the prognosis of BRD. Similar results investigating epidemiological associations with mortality risk were also observed by Babcok and colleagues [27]. Cattle treated later in the feeding phase (mid DOF and late DOF) showed a greater probability of both FTF and DNF, suggesting that cattle in these stages may be at increased risk due to cumulative stressors and more advanced disease progression.

Similarly, cattle with a BW < 272 kg also displayed a higher probability of DNF and FTF. This could be attributed to the fact that lighter cattle may be more vulnerable to BRD due to their transport and adaption stressors. Increased mortality risk in lighter cattle has been observed in multiple reports [24,28,29].

### 4.3. Ultrasound Lung Score, B-Line Count, and Moth Sign

The TT-POCUS variables were found to be strongly associated with both outcomes of interest. The probability of DNF and FTF was significantly higher in cattle with ULSs of 5, with a ULS of 4 also demonstrating a greater probability for FTF compared to lower scores. B-line count was another important factor, with cattle showing ≥3 B-lines having an elevated likelihood of both DNF and FTF. The presence of the moth sign, a pleural abnormality observed during ultrasound, further increased the probability of poor outcomes. These results support the utility of TT-POCUS in assessing lung pathology and guiding prognosis in BRD cases.

Interestingly, other TT-POCUS variables, such as A-line count and pleural effusion, were not significantly associated with DNF or FTF. This suggests that while certain ultrasonographic findings are highly relevant for establishing a prognosis, others may have limited predictive value for these specific outcomes. The use of ULSs, moth sign and B-line count as prognostic markers in chute-side settings provides a practical approach to help veterinarians to make informed management decisions, enabling more targeted treatment and potentially improving animal welfare and feedyard productivity.

These findings align with previous research suggesting that severe lung consolidation is indicative of poorer outcomes in cattle with respiratory disease [20,30,31,32,33]. In 2019, Timsit and collaborators showed that cattle with lung consolidation (≥5 cm) at the first diagnosis of bronchopneumonia had a significantly higher risk of treatment failure compared to cattle with less severe lung lesions [20]. Another similar finding was reported by Rademacher and colleagues in 2014, where mortality risk was increased when lung consolidation was present [21]. B-line count and moth sign (sub-pleural consolidation and pleural thickening) are often described in POC evaluations in small animals, where the presence of moth sign and a greater number of B-lines is linked to greater pulmonary insult [25]. In human medicine, B-line evaluation is the core artifact POC assessed for pulmonary diagnostics [34,35,36,37,38,39,40,41,42].

### 4.4. Pulse Oximetry and Pulmonary Auscultation

Pulse oximetry and pulmonary auscultation have been reported successfully in cattle due to their viability and association with lung disease [11,12,13,14,15]. Although pulse oximetry and pulmonary auscultation were included as part of the chute-side evaluations, their parameters were not significantly associated with either DNF or FTF in the final models. Blood oxygen saturation (SPO2) and pulse per minute (PPM) values showed no clear differences between recovered and DNF or FTF and FTS cases, and similarly, cranioventral and caudo-dorsal pulmonary auscultation scores (PASs) were not associated with these outcomes. The lack of significant associations may be attributed to several factors, including the variability in SPO2 readings based on sensor placement (external ear), which was a different placement than in previous research [11,12,13], or the potential for subclinical lung pathology not detectable through auscultation.

Further research is warranted to refine the use of TT-POCUS and other POC diagnostics in feedyard settings. Future studies could explore the development of standardized scoring systems for ultrasound findings and evaluate the cost-effectiveness of implementing POC diagnostics as part of routine BRD management. Additionally, investigating the interplay between multiple POC parameters and their combined predictive power could enhance the accuracy of prognosis in BRD cases.

## 5. Conclusions

The identification of TT-POCUS variables as significant prognostic indicators has practical implications for feedyard management. By providing real-time, actionable data at the time of first treatment, TT-POCUS allows for early identification of cattle at higher risk for FTF and DNF. This can support more informed decision-making, including treatment adjustments, selective culling, and resource allocation, ultimately enhancing animal welfare and reducing economic losses.

## Figures and Tables

**Figure 1 vetsci-12-00369-f001:**
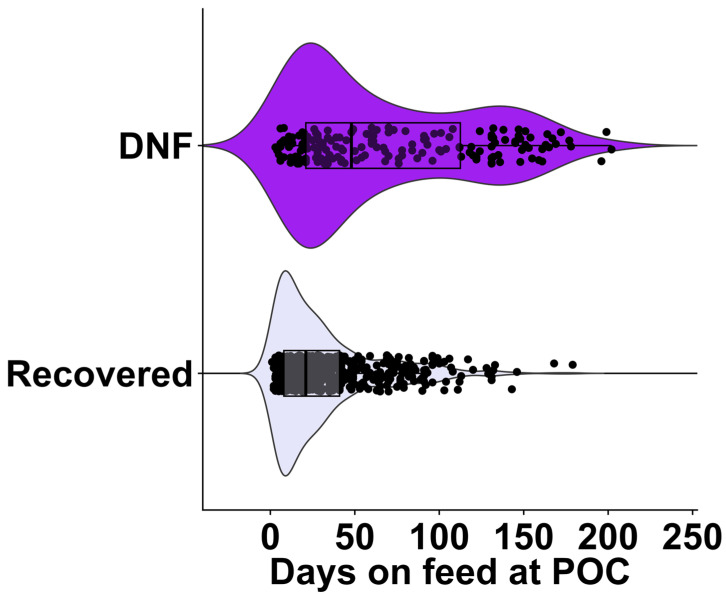
The raincloud boxplot displays temporal distribution of did-not-finish outcome model on the days on feed at time of chute-side evaluations (POC) on individuals enrolled in this study (n = 819). It depicts the 60-day outcomes for DNF = did not finish (culled/dead) and recovered animals that were alive at the 60-day post-evaluation. The boxplot represents the upper and lower quartiles, and the line within the box represents the median.

**Figure 2 vetsci-12-00369-f002:**
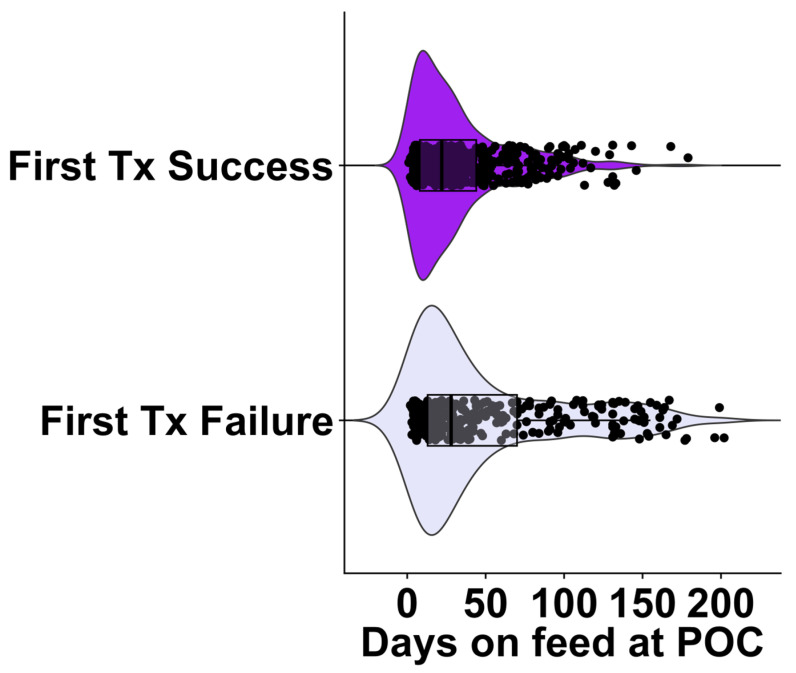
The raincloud boxplot displays temporal distribution of the treatment success outcomes model on the days of feed ingestion at time of chute-side evaluations (POC) on individuals enrolled in this study (n = 819). The 60-day outcomes of FTF = first treatment failure (re-treat/cull/death) and FTS = first treatment success; animals that were alive at the end of the 60-day post-evaluation period and were not re-treated. The boxplot represents the upper and lower quartiles, and the line within the box represents the median.

**Figure 3 vetsci-12-00369-f003:**
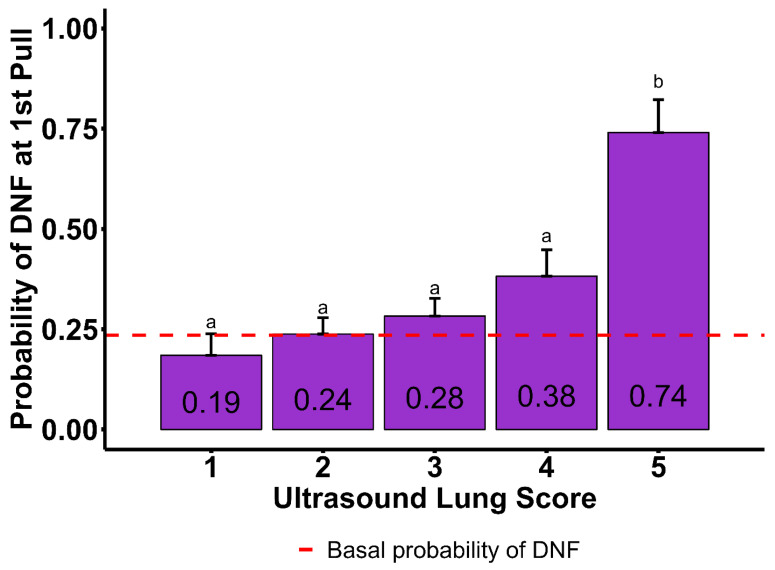
Model-adjusted probability of did not finish (cull/death; DNF) rate by ultrasound lung score (ULS) from a population of 819 cattle evaluated with targeted thoracic point-of-care ultrasound (TT-POCUS). The prevalence of DNF in this studied population was 23%. The ULSs are defined as follows: 1—the absence of B-lines or less than 2 thin B-lines present; 2—the presence of 3 or more thin B-lines; 3—the presence of merged B-lines; 4—the presence of several wide/merged B-lines, abnormal pleural (moth sign); 5—consolidated lung, might be accompanied by moth sign and effusion based on description of lung injury. ^a,b^ The ULSs that do not share the same superscript display evidence of significant difference (*p* < 0.05).

**Figure 4 vetsci-12-00369-f004:**
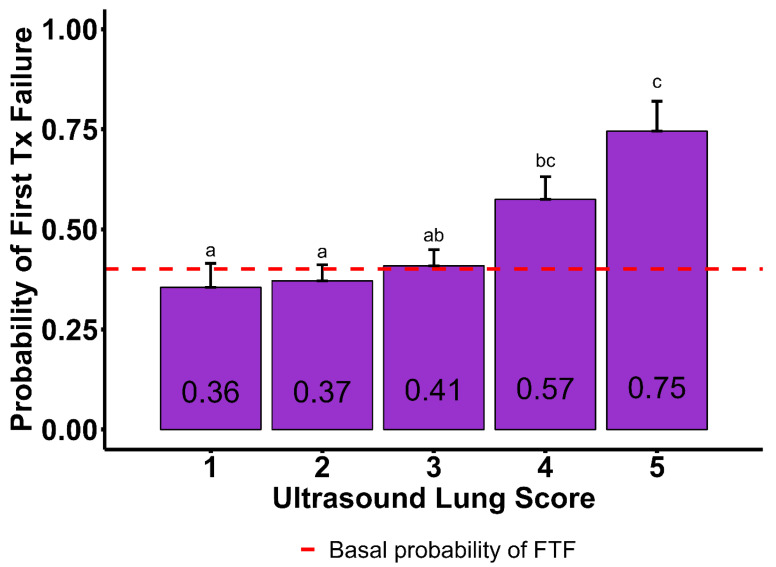
Model-adjusted probability of first treatment failure (re-treatment/cull/death; FTF) by ultrasound lung score (ULS) from a population of 819 cattle evaluated with targeted thoracic point-of-care ultrasound (TT-POCUS). The prevalence of FTF in the studied population was 40%. ULSs are defined as follows: 1—absence of B-lines or less than 2 thin B-lines present; 2—presence of 3 or more thin B-lines; 3—presence of merged B-lines; 4—presence of several wide/merged B-lines, abnormal pleural (moth sign); 5—consolidated lung; might be accompanied by moth sign and effusion based on description of lung injury. ^a,b,c^ The ULSs that do not share the same superscript display evidence of significant difference (*p* < 0.05).

**Table 1 vetsci-12-00369-t001:** Descriptive statistics of did-not-finish (DNF) outcome model cattle according to their demographics (sex, weight, and days on feed categories), real-time chute-side evaluations (ultrasound lung score, blood oxygen saturation, pulse, auscultation scores, A-line count categories, B-line count categories, pleura effusion, moth sign); and post hoc measurement (B-line area).

	60-Day Outcome	
	Recovered ^1^	DNF ^2^
**Sex**		
Heifer	348	126
Steer	280	65
**Days on feed**		
0 to 42	481	92
43 to 71	77	27
>71	70	72
**Bodyweight, average (SD), kg**	353.1 (±73.9)	389.9 (±115.2)
**Bodyweight, kg**		
<272	63	27
272 to 361	324	68
362 to 453	188	36
>453	53	60
**Real-time measurements**		
**Ultrasound lung score, head ***		
1	78	15
2	245	52
3	224	61
4	67	39
5	14	24
**Blood oxygen saturation (SD), % *^,§^**	85.7 (±8.5)	83.1 (±9.4)
**Pulse, average (SD) *^,§^**	74.5 (±22.2)	75.9 (±25.1)
**Auscultation score (cranioventral), head *^,§^**		
1	12	1
2	103	9
3	137	12
4	89	20
5	39	21
**Auscultation score (caudo-dorsal), head *^,§^**		
1	103	6
2	129	13
3	83	14
4	54	19
5	11	11
**A-line count, head ***		
0 to 2	243	117
≥3	389	70
**B-line count, head ***		
0 to 2	482	67
≥3	150	120
**Pleural effusion, head ***		
Absent	530	134
Present	102	53
**Moth sign ^†^, head ***		
Absent	491	77
Present	141	110
**Post hoc measurements ^‡^**		
**B-line area, cm^2^, average (SEM)**	16.11 (±13.1)	19.7 (±18.2)

^1^ Animals that were alive for the 60-day post-evaluation period; ^2^ animals that were culled or dead within the 60-day post-evaluation period. * Variables collected at chute-side real time (n = 819). ^†^ Moth sign is noted when the pleural line shows irregularities like pleural thickening and sub-pleural consolidation (indentations). ^‡^ Post hoc measurement (n = 710) was performed using ImageJ 1.54. Measurements were taken from a still image, where the frame of choice was the frame with most ultrasonographic artifacts and abnormalities. ^§^ Only animals enrolled in the fall (n = 443) were evaluated with pulse oximetry and pulmonary auscultations.

**Table 2 vetsci-12-00369-t002:** Descriptive statistics of first treatment outcome model, cattle (according to their demographics (sex, weight, and days on feed categories)), real-time chute-side evaluations (ultrasound lung score, blood oxygen saturation, pulse, auscultation scores, A-line count categories, B-line count categories, pleura effusion, moth sign); and post hoc measurement (B-line area).

	First Treatment 60-Day Outcome	
	Failure ^1^	Success ^2^
**Sex**		
Heifer	205	269
Steer	122	223
**Days on feed**		
0 to 42	210	363
43 to 71	36	68
>71	81	61
**Bodyweight, average (SD), kg**	363.9 (±100.4)	360.1 (±75.8)
**Bodyweight, kg**		
<272	43	47
272 to 361	156	236
362 to 453	65	159
>453	63	50
**Real-time measurements**		
**Ultrasound lung score, head ***		
1	30	62
2	101	197
3	108	177
4	61	45
5	27	11
**Blood oxygen saturation (SD), % *^,§^**	84.1 (±9.2)	85.9 (±8.3)
**Pulse, average (SD) *^,§^**	75.1 (±23.2)	74.6 (±22.3)
**Auscultation score (cranioventral), head *^,§^**		
1	4	9
2	34	78
3	39	110
4	42	67
5	31	29
**Auscultation score (caudo-dorsal), head *^,§^**		
1	27	82
2	39	103
3	35	62
4	34	39
5	15	7
**A-line count, head ***		
**0 to 2**	181	179
**≥3**	137	322
**B-line count, head ***		
0 to 2	163	386
≥3	156	114
**Pleural effusion, head ***		
Absent	245	419
Present	74	81
**Moth sign ^†^, head ***		
Absent	166	403
Present	152	98
**Post hoc measurements ^‡^**		
**B-line area, cm^2^, average (SEM)**	18.9 (±17.3)	15.4 ± 12.2

^1^ Animals that were living at the end of the 60-day post-evaluation; ^2^ animals that were culled or dead within 60-day post-evaluation; * variables collected at chute-side in real time (n = 819). ^†^ Moth sign is noted when the pleural line shows irregularities like pleural thickening and sub-pleural consolidation (indentations); ^‡^ Post hoc measurement (n = 710) was performed using ImageJ 1.54. Measurement was taken from a still image, where the frame of choice was the frame with most ultrasonographic artifacts and abnormalities. ^§^ Only animals enrolled in the fall (n = 443) were evaluated with pulse oximetry and pulmonary auscultations.

**Table 3 vetsci-12-00369-t003:** Probabilities of variables that were associated with the outcome of interest (did not finish or first treatment failure) converted from log-odds generated from multivariate logistic regression models grouped by specific model.

60-day Outcome Models *						
	Did Not Finish ^1^			First Treatment Failure ^2^		
Items	Probability	SE	*p*-Value	Probability	SE	*p*-Value
**Sex**						
Heifer	0.402	0.047		0.542	0.041	
Steer	0.322	0.052	0.223	0.483	0.051	0.235
**Days on feed**						
0 to 42	0.178	0.030		0.433	0.042	
43 to 71	0.319	0.069	0.018	0.437	0.068	0.983
>71	0.639	0.057	0.001	0.662	0.058	0.004
**Bodyweight, kg**						
<272	0.612	0.102		0.603	0.079	
272 to 361	0.337	0.062	0.005	0.535	0.053	0.397
362 to 453	0.247	0.048	0.001	0.385	0.047	0.008
>453	0.477	0.076	0.324	0.527	0.065	0.444
**Ultrasound Lung Score**						
1	0.185	0.053		0.355	0.060	
2	0.237	0.040	0.402	0.371	0.040	0.798
3	0.283	0.043	0.146	0.408	0.040	0.416
4	0.382	0.065	0.017	0.574	0.056	0.006
5	0.740	0.081	0.001	0.745	0.074	0.001
**B-line**						
0 to 2	0.242	0.040		0.430	0.042	
≥3	0.605	0.058	0.001	0.638	0.046	0.001
**Moth sign**						
Absent	0.2375	0.037		0.371	0.040	
Present	0.506	0.053	0.001	0.652	0.043	0.001

^1^ Animals that were alive at the end of the 60-day post-evaluation; ^2^ animals that were culled or died within the 60-day post-evaluation period; * Two distinct logistic regression models were used to assess the two different outcomes: treatment success outcome (n = 819), and the latter for the did not finish outcome (culled or dead) (n = 819). The log-odds from the logistic regression models were converted to probabilities using the package “emmeans” in RStudio.

## Data Availability

The original contributions presented in this study are included in the article. Further inquiries can be directed to the corresponding author.

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
