# Peer review of "Associations Between Thoracic Ultrasound Chute-Side Evaluations and 60-Day Outcomes in Feedyard Cattle at Time of First Treatment for Respiratory Disease"

_vetsci, 2025, doi:10.3390/vetsci12040369_

Round 1

Reviewer 1 Report

Comments and Suggestions for Authors

Line 98: which were the criteria for re-treatment?

Information about which was the first treatment should be provided

Tables: which is the meaning of “head”?

Line 347: Please correct this sentence

Lines 400-403: Could lower weight be associated with a chronic respiratory disease?

Author Response

The authors sincerely thank the reviewer for the detailed comments and valuable suggestions.

Reviewer 1:
Comment 1: Line 98: which were the criteria for re-treatment?

Information about which was the first treatment should be provided

Thanks for your feedback. The authors addressed this concern in lines 99-101.

Comment 2: Tables: which is the meaning of “head”?

Thanks for your question. Head is a common term in the cattle industry, meaning one bovine or animal. Since it is a common term in the industry, I will refer to the editor if you like me to change this term (head) to something else.
Thanks

Comment 3: Line 347: Please correct this sentence

Thanks for your feedback. The authors addressed this concern in lines 347-348.

Comment 4: Lines 400-403: Could lower weight be associated with a chronic respiratory disease?

Thanks for your comment. The authors addressed this in Lines 402-405.

Reviewer 2 Report

Comments and Suggestions for Authors

Dear authors,

The topic of the present document is very important for a lot feedlot responsible people around the world, so, please, check all comments included into the document and improve the format of the main ideas to get more clear the data and information.

Comments on the Quality of English Language

I can not native knowledge and/or authority to suggest English expression

Author Response

Dear authors,

The topic of the present document is very important for a lot feedlot responsible people around the world, so, please, check all comments included into the document and improve the format of the main ideas to get more clear the data and information.

We thank the reviewer for the thoughtful feedback and constructive insights, which greatly improved the manuscript.

Comment 1:
Please, I not sure, but the final dot could be post parenthesis....??? If, is rigth, all of them.

Thanks for catching this formatting mistake. In-text citations corrected throughout the manuscript.

Comment 2:
Please, a long of the results section, there are a lot abbreviation & description of the same physiological parameters. Ex. PPM. Simplify

Line 229: Thanks for this great suggestion to enhance clarity of this manuscript. Changes were made to enhance the text. 

Comment 3:
Please, a long of the results section, there are a lot abbreviation & description of the same physiological parameters. Ex. PPM. Simplify.

Results section: Thanks for the suggestion to enhance the document’s clarity. Author’s removed excess description and fixed the overly repetitive use of acronyms.

Comment 4:

Line 224: Highlighted text (cm2)

Thanks for the suggestion. Cm2 was fixed to cm2 throughout the manuscript.

Comment 5:

Line 297: Highlighted text

Line 300: Fixed the punctuation, thanks.

Comment 6:
Line 299: Highlighted text

Line 302: Fixed the punctuation, thanks.

Comment 7:

Line 302-311: Strikethrough Text (%)

Lines 302-315: Thanks for the correction. Fixed the formatting to display the results

Comment 8:
Line 344,346,347: Highlighted text

Line 347-350: Fixed the punctuation and the reference to Table 3. Thanks.

Line 349-359: Strikethrough Text (%)

Lines 302-315: Thanks for the correction. Fixed the formatting to display the results.

Comment 8:
Line 344,346,347: Highlighted text

Line 347-350: Fixed the punctuation, and the reference to table 3. Thanks.

Comment 9:
Line 357: Highlighted text

Line 360: Fixed the reference to Figure 4. Thanks.

Comment 10:
Lines 376-379: If full description of the variable (did not finish) delete DNF. One or the other.
Lines 378-380: Thanks for the correction. The authors comply with this suggestion.

Reviewer 3 Report

Comments and Suggestions for Authors

Between Chuteside Evaluations and 60-day Outcomes in 2 Feedyard Cattle at Time of First Treatment for Respiratory Dis- 3 ease

Keywords: Chuteside diagnostics, feedyard cattle health, precision veterinary medicine, 37 bovine respiratory disease, thoracic ultrasound.

All words in the title are keywords. Therefore, words from the title should not be repeated as keywords. Authors should remove repeated words from one of these locations!

This does not fit in the conclusion. Perhaps the authors could suggest this at the end of the discussion. As the last paragraph. But not in the conclusion. “However, further research is warranted to refine the use of TT-POCUS and other 453 POC diagnostics in feedyard settings. Future studies could explore the development of 454 standardized scoring systems for ultrasound findings and evaluate the cost-effectiveness 455 of implementing POC diagnostics as part of routine BRD management.”

The job is good, well written and well analyzed.

It should be published!

Author Response

We genuinely appreciate the reviewer's critical evaluations and insightful recommendations.

Comment 1:
Keywords: Chuteside diagnostics, feedyard cattle health, precision veterinary medicine, 37 bovine respiratory disease, thoracic ultrasound. All words in the title are keywords. Therefore, words from the title should not be repeated as keywords. Authors should remove repeated words from one of these locations!

Thank you for your insightful suggestion. Keywords were replaced to enhance the searchability of this manuscript in Lines 37-38.

Comment 2:
This does not fit in the conclusion. Perhaps the authors could suggest this at the end of the discussion. As the last paragraph. But not in the conclusion. “However, further research is warranted to refine the use of TT-POCUS and other 453 POC diagnostics in feedyard settings. Future studies could explore the development of 454 standardized scoring systems for ultrasound findings and evaluate the cost-effectiveness 455 of implementing POC diagnostics as part of routine BRD management.”

Thank you for your great suggestion. The mentioned text (in your comment) was moved to the end of the discussion and reads much better now. Lines 447-452.